# Efficacy of Emu Oil Transfersomes for Local Transdermal Delivery of 4-OH Tamoxifen in the Treatment of Breast Cancer

**DOI:** 10.3390/pharmaceutics12090807

**Published:** 2020-08-25

**Authors:** Usha Sundralingam, Srikumar Chakravarthi, Ammu Kutty Radhakrishnan, Saravanan Muniyandy, Uma D. Palanisamy

**Affiliations:** 1School of Pharmacy, Monash University Malaysia, Jalan Lagoon Selatan, Bandar Sunway, Selangor Darul Ehsan 47500, Malaysia; usha.sundralingam@monash.edu; 2Jeffrey Cheah School of Medicine and Health Sciences, Monash University Malaysia, Jalan Lagoon Selatan, Bandar Sunway, Selangor Darul Ehsan 47500, Malaysia; ammu.radhakrishnan@monash.edu; 3Faculty of Medicine, Bioscience and Nursing, MAHSA University, Jalan SP 2, Bandar Saujana Putra, Jenjarom Selangor 42610, Malaysia; srikumar@mahsa.edu.my; 4Fatima College of Health Sciences, Al Maqam, Al Ain 24162, UAE; drmsaravanan@gmail.com

**Keywords:** local transdermal therapy, tamoxifen, 4-hydroxytamoxifen, DCIS, emu oil

## Abstract

Oral tamoxifen used in the prevention and treatment of ductal carcinoma in situ (DCIS) (estrogen-positive) patients has limited acceptance, due to its adverse side effects. The efficacy of tamoxifen is related to its major metabolite, 4-hydroxytamoxifen. Local transdermal therapy of 4-hydroxytamoxifen to the breast might avert the toxicity of oral tamoxifen, while maintaining efficacy. We aim to study the skin irritancy, as well as to evaluate the efficacy of the developed transfersome formulations, with/without emu oil, using a syngeneic mouse model of breast cancer. We also quantified tamoxifen/4-hydroxytamoxifen concentrations in blood plasma and performed histopathology. The skin irritancy test showed that the pure emu oil and transfersome formulations with or without the emu oil did not cause skin irritancy in the animals studied. A sensitive and specific LC–MS/MS method for the quantification of tamoxifen and 4-hydroxytamoxifen was developed and validated. Studies on tumor volume and necrosis (histopathology) using the breast cancer mouse model showed that the 4-OHT transfersomal formulations, with and without emu oil, showed comparable efficacy with that of orally administered tamoxifen. However, the transfersomal formulations, with and without emu oil, resulted in significantly lower (10.24 ± 0.07 and 32.45 ± 0.48 ng/mL, respectively) plasma concentrations of 4-hydroxytamoxifen, compared to the oral tamoxifen (TAMX) group (634.42 ± 7.54 ng/mL). This study demonstrated the potential use of emu oil in a local transdermal formulation for the treatment of breast cancer and its reduced adverse effects.

## 1. Introduction

Breast cancer (BC) is the most common cancer among women; impacting over two million women each year. In 2018, it was estimated that 627,000 women died from BC, which correspond to approximately 15% of cancer-related deaths among women. In Malaysia, the incidence, mortality, and prevalence of BC was reported to be the highest among other cancers; where 7593 new cases were reported in 2018 [1].Population-based BC screening programs found that ductal carcinoma in situ (DCIS) represents 20–25% of all BC, and from statistical analysis it is predicted that 1 in 33 women are likely to be diagnosed with DCIS of the breast during their lifetime [2]. DCIS is a non-invasive BC that starts inside the milk ducts in the breast tissue. It is considered to be non-invasive because it normally does not spread into any of the normal surrounding breast tissues. Although DCIS is not life-threatening, it increases the risk of developing invasive BC later on. To date, the clinical significance of a DCIS diagnosis and optimal approaches to treatment are still topics of uncertainty and concern for patients and clinicians [3,4]. Women with DCIS that are estrogen receptor (ER)-positive can be given hormone therapy with tamoxifen (TAMX), which is associated with decreased risk of future breast events, including invasive cancer and DCIS in unaffected breast [5,6,7]. Currently, TAMX is administered through oral and parenteral routes. Oral TAMX is a pro-drug, which is converted to its metabolite 4-hydroxytamoxifen (4-OHT) through cytochrome P450, one of the phase I drug metabolizing enzymes [8]. The 4-OHT is reported to have better binding affinity to ER, compared to TAMX. Oral administration of TAMX is proven to be quite effective. However, it can cause certain side-effects like distaste for food, abdominal cramps, nausea, and vomiting. A few patients also experience some in-frequent side-effects, such as endometrial carcinoma, ocular problems, and thromboembolic disorders and acquired drug-resistance on long-term therapy [9,10,11,12].

Diagnosis and treatment of cancers have ongoing physical and psychological effects on the life and well-being of patients, which remain an issue in long-term survival [13]. Current research in mitigating BC failed to address the issue of patient compliance. Studies showed that reducing and managing this disease and any symptoms related to its treatment could improve the quality of life of women with BC [14]. Therefore, there is a need to look for alternative modes of treatment, which can improve the bioavailability or efficacy of the drugs used for treatment, which have reduced systemic effects and are less expensive. These features can help to improve the quality of life of BC patients, which will consequently improve their compliance. One possible solution is to use the local transdermal delivery approach, to directly administer the active drug (4-OHT) to the affected area on the breast. It is hypothesized that due to the embryological origins of the breast as a skin appendage (a modified eccrine gland), a well-developed internal lymphatic circulation can promote accumulation of drug in the breast [15]. Treatment of BC using topical formulations is still at its infancy. Nevertheless, local transdermal treatment of BC appears to be promising, as it can help to avoid some of the systemic side-effects observed with the oral or parenteral routes. This provides an improved experience for the patients, which in turn could make them be more compliant to their treatment; consequently, improving the course of the disease. Recently, there was a couple of randomized control trials (RCT) on the use of a transdermal-based drug delivery system for the treatment of BC [16]. This study showed that transdermal therapy using a 4-OHT gel was safer than oral tamoxifen. However, the potential use of a topical transdermal therapy for the treatment of BC is not fully endorsed by healthcare professionals, due to the limited number of clinical studies carried out.

We developed a highly deformable vesicle formulation (Transfersomes™), which was shown to have the ability to permeate through the pores on skin, which are one-tenth of their own diameter when applied under non-occlusive conditions [17]. Transfersomes are morphologically similar to liposomes, but they differ in function as they traverse intact skin, carrying therapeutic amounts of drugs into the systemic circulation [18,19,20]. Unlike the conventional liposomes, where permeation is only limited to the outer layers of the stratum corneum (SC), these highly deformable vesicles are able to permeate through the pores of the SC that are one-tenth of their own diameter, when applied under non-occlusive conditions [17].

The transfersomes were formulated with and without emu oil. Emu oil, derived from the Emu (*Dromaius novaehollandiae*), was traditionally used by Australian aboriginals to facilitate wound healing and alleviate pain from musculoskeletal disorder. More recently, the oil is extensively used in the cosmetics and pharmaceutical industry. Emu oil consists of polyunsaturated fatty acids (PUFA), Omega 3, 6, 9, essential fatty acids (EFA), vitamins, and amino acids, which help maintain the integrity of the skin membrane [21]. The presence of omega-3 and omega-9 fatty acids are not only beneficial for reducing hypertension, stroke risk, decreasing risk of arthritis, but most of all, it aids in the prevention of cancer and plays a role in inhibiting breast cancer and promoting healthy inflammation responses [22]. A recent study proved emu oil to exhibit anti-osteoclastic and osteotrophic properties [23]. A few clinical trials identified the use of emu oil in vulvar pain relief in women [24], prevention of radiation dermatitis [25], and comparing its efficacy between clotrimazole and hydrocortisone in the treatment of seborrheic dermatitis [26]. Hence, the ability of emu oil to act as a penetration enhancer as well as its anti-inflammatory property makes it an ideal agent to be used in this transfersomal formulation.

In line with this, the physical–chemical characterization and ex vivo permeability studies of the developed optimum transfersome formulations with soy phosphatidyl choline (SPC) and sodium taurocholate (NaTC) [(85:15%, *w*/*v*)], prepared in the presence or absence of emu oil is established and we already reported this (Figure 1) [27]. In the present study, we report the results from skin irritancy studies, as well as the results of the evaluation efficacy of the transfersome formulations with emu oil, using a syngeneic mouse model of breast cancer [28] and quantification of TAMX/4-OHT concentration in blood plasma and histopathology.

## 2. Materials and Methods

### 2.1. Animals (Mice)

Forty-five (45) 6-week old female BALB/c mice (15–20 g) were obtained from the animal holding facility of Monash University, Malaysia. The animals were housed under standardized animal facility and husbandry conditions with a room temperature of 22 °C, relative humidity of 55%, and food and water available ad libitum. The mice were kept in autoclavable propylene cages with corn-on-cob bedding, with six mice per cage. The cages were cleaned weekly. All experiments were carried out with the approval of the Animal Ethics Committee (AEC) of Monash University, Australia (MARP/2015/066); in accordance with the 2004 Australian Code of Practice for the Care and Use of Animals for Scientific Purposes and relevant Victorian legislation, 7 August 2015.

### 2.2. Cell Line

The 4T1 murine mammary cancer cell line was purchased from the American Type Culture Collection ((ATCC, Manassas, VA, USA (Catalogue number: CRL-2539™)). These ER positive cells were maintained in a culture medium recommended by the ATCC. The culture medium used was Dulbecco’s Modified Eagle Medium (DMEM) (contains non-essential amino acids, 4 mM L-glutamine, 4500 mg/L glucose, and 1500 mg/L sodium bicarbonate), supplemented with 10% fetal bovine serum (FBS) (Biowest, Nuaille, France), 1% HEPES (1M) (Invitrogen, Carlsbad, CA, USA) and 1% penicillin streptomycin solution (units/mL penicillin and 10,000 μg/mL streptomycin) (Invitrogen, Carlsbad, CA, USA). The monolayer cell cultures were incubated at 37 °C, in a humidified atmosphere of 5% CO_2_. The culture medium was replenished every 4–5 days. Cells between the 10th and 15th passages were used for the induction of breast cancer in mice.

### 2.3. Test Drugs

The drugs, tamoxifen (Catalogues No. 06734) and (Z)-4-hydroxytamoxifen (Catalogues No. H6278) were purchased from Sigma Aldrich, St. Louis, MO, USA.

### 2.4. Chemicals

Pharmaceutical grade pure Emu oil was purchased from ClinicReady, Queensland, Australia. The fatty acid composition of the emu oil consists mainly of oleic (51.6%) and palmitic (22.5%) acid. Dichloromethane, Acetonitrile, Ethanol, Isopropy Alcohol (IPA), and DMSO were supplied by Mallinckrodt Chemicals, Bedminster, NJ, USA. All other chemicals and reagents used in the study were of analytical grade and purchased of commercial sources.

### 2.5. Transfersome Formulation

Transfersomes were prepared using the thin-film hydration technique adapted from previously established methods [29,30]. In brief, the formulation consists of soy phosphatidyl choline, edge activator, 1% (*w*/*v*) emu oil, and test drug (4OHT) (1 mg/mL). The two formulations used in this study were transfersome formulations of 4-OHT in SPC and sodium taurocholate (NaTC) (85:15%, *w*/*v*) (DLT (85/15) in the presence of emu oil (DLT + EO (85/15)), which was described previously [27]. The percentage drug entrapment efficiency of the formulations after removal of free drug were 95.1 ± 2.70 and 72.3 ± 0.15%, respectively.

### 2.6. Skin Irritancy Studies

The skin irritation potential of emu oil, unloaded transfersomes, and DLT + EO (85:15; *w*/*v*) transfersomes were studied on the BALB/c female mice, using the Draize method [31]. In brief, the animals were randomly assigned into four groups (A to D) of three mice each (*n* = 3 per group) in separate polypropylene cages. Animals in group A served as the control group. Animals in groups B and C were treated daily with topical application of 100 µg of unloaded transfersomes (Group B) and emu oil (Group C); whilst animals in group D were treated daily with 100 µg of 4-OHT (1 mg/mL/day) transfersome with emu oil (DLT–EO). Hair from the breast area was removed using hair removal cream (Veet^®^), 24 h before the first application, and as necessary during the duration of the study (at least 4 h before application of the test formulations). Treatment was carried out for 10 days on a 2 cm^2^ surface area on the skin in the breast area of the mice. Animals were examined daily for signs of skin irritation, prior to application of the transdermal formulation and throughout the treatment period. The “irritation score” as defined by Draize [31] (0 = no differences, 1 = light erythema, 2 = well defined erythema, 3 = strong erythema and 4 = very strong erythema with presence of a scar) was used to record the degree of irritation. The mice were housed under standardized environmental conditions throughout the study. At the end of the 10th day, the animals were humanely sacrificed. Skin biopsies (1 cm^2^) taken from all treated areas from all animals were preserved in 10% (*w*/*v*) buffered-formalin solution. These tissues were processed for the histopathological studies. Tissue sections were stained by hematoxylin and eosin (H&E) using the standard staining procedure [32] and were examined using a light microscope (Olympus, Tokyo, Japan). Any test substances found to cause irritation with a Draize irritation score of 2 and above were not tested in the efficacy studies.

### 2.7. Efficacy of Topical Application of 4-Hydroxytamoxifen (4-OHT) Transfersomal Formulations

About 10^4^ 4T1 tumor cells were orthotopically injected into the mammary gland (second nipple) of 6-weeks old female mice [33]. When the tumor was palpable (average diameter of 3–4 mm; about 65 mm^3^ in volume) around 11–12 days post-inoculation [33], the mice were randomly assigned in to one of five study groups. Each group had six mice (*n* = 6 per group) that were housed in separate polypropylene cages. The mice were maintained under standardized environmental conditions, throughout the study. Mice in group A served as the control group, whilst mice in group B were administered with oral TAMX. The mice in the remaining three groups were treated daily with a topical application of 100 µg of emu oil (Group C), 4-OHT transfersome with emu oil (DLT–EO) (Group D) or 4-OHT transfersome (DLT) (Group E). Prior to applying the transdermal formulations, hair was removed from the breast area using hair removal cream, as described above. The various transdermal preparations were applied directly on 2 cm^2^ surface area on the skin of around the affected breast for 18 days. Each mouse received 120 µg (0.6 mg/kg) of oral TAMX or 100 µg (1 mg/mL/day) of topical 4-OHT. The perpendicular diameters, i.e., the length (L) and the width (W) of the tumor in each mouse were measured every 2 to 3 days, using a digital calliper. Tumor volume was calculated using the following formula [19]:V = 0.52 × L^2^ × W(1)

At the end of the 18 days of treatment, the mice were humanely euthanized. They were placed in a jar with ethyl ether, until unconscious, and blood was withdrawn via a cardiac puncture. At autopsy, various organs (e.g., liver, lung, brain, and tumor) were harvested for histopathology studies. The harvested organs were fixed in 10% (*v*/*v*) neutral buffered formalin. On evaluating the efficacy of the formulation(s), the parameters studied included tumor volume, quantification of 4-OHT, and TAMX in blood plasma, and histopathological studies.

### 2.8. Quantification of Tamoxifen (TAMX) and Its Metabolite 4-OHT in Mice Plasma

Blood samples were obtained via a cardiac puncture, 24 h after the last administration/application of the treatments. Blood was collected in 1.5 mL microcentrifuge tube containing ethylenediaminetetraacetic acid (EDTA), which served as the anticoagulant. The tubes were immediately mixed and centrifuged at 1300× *g* for 10 min at 4 °C. Plasma from each sample was transferred into a clean microcentrifuge tube and stored at −80 °C, prior to analysis.

Stock solutions containing 0.5 mg/mL TAMX and 4-OHT were prepared in methanol and stored at −80 °C. These stock solutions were used to prepare the working stock solution (8 µg/mL) of TAMX or 4-OHT. The primary calibration curves (500–0.5 ng/mL) for TAMX and the 4-OHT standards were constructed using freshly prepared working stock solutions. Quality control (QC) samples were prepared at low, medium and high concentration levels of 1 ng/mL (QC low), 20 ng/mL (QC medium), and 200 ng/mL (QC high). For high performance liquid chromatography (HPLC analysis), 100 µL of the internal standard working solutions (8 ng/mL each) (containing both TAMX and 4-OHT) and 100 µL of plasma from each mouse were added to 200 µL of methanol/0.2 M HCl (1:1, *v*/*v*) in 1.5 mL microcentrifuge tubes, and vigorously vortexed for 30 s. Then, the samples were centrifuged at 1320× *g* for 10 min at 4 °C. Subsequently, the supernatant was transferred into another 1.5 mL microcentrifuge tube and centrifuged (1320× *g* for 10 min at 4 °C), to ensure complete removal of precipitated proteins. The supernatants were transferred into HPLC vials, and placed into the chilled (4 °C) auto sampler, from which aliquots of 1 µL were injected into the LC column.

The LC–MS/MS method was validated in agreement with the Guidance for Industry, Bioanalytical Method Validation, as specified by the FDA [34]. In order to establish the specificity of the method, blank BALB/c mice plasma samples of three different lots were analyzed to determine the potential endogenous contaminating compounds that might interfere with the assay. Linearity of the calibration curve was evaluated by linear regression analysis, which was calculated by the least square regression analysis. The slope and other statistics of calibration curves were calculated by linear regression and analysis of variance (ANOVA). All validation runs were performed in triplicates on three consecutive days, to assess inter-day and intra-day variation (*n* = 9). The intra-day assay precision and accuracy were estimated by analyzing the QC samples. The inter-day assay precision was determined by analyzing the three levels of QC samples on three different runs. Accuracy was assessed by comparing the calculated mean concentrations with the known concentration. The QC samples were analyzed against the calibration curve. The standard deviation (SD) and relative standard deviation (RSD) were calculated from the QC samples, and used to estimate the intra- and inter-day precision. The criteria for acceptability of the data included accuracy and precision within ±15% from the nominal values [34]. The lower limit of detection (LLOD) was defined as the lowest detectable concentration, taking into account a signal-to-noise ratio of 3. Lower limit of quantification (LLOQ) was determined as the lowest concentration at which the precision (RSD) was <20% with a signal-to-noise ratio of 10.

Estimation of TAMX and 4-OHT in plasma of mice was carried out using the Agilent reverse phase C-18 ZORBAX SB narrow-bore column (2.1 × 150 mm) of 3.5 µm, thermostatted at 25 °C. The TAMX and 4-OHT were isolated at the flow rate of 0.6 mL min^−1^, using the mobile phase consisting of water (mobile phase A) and acetonitrile (mobile phase B); both acidified with 0.1% (*v*/*v*) formic acid. A linear gradient separation was used with 60% (B) from 0 to 0.3 min, 60–65% (B) from 0.3 to 1.6 min, followed by 65–80% (B) from 1.6 to 3 min, and finally 80–100% (B) from 3.0 to 3.1 min. An autosampler set at 4 °C injected volumes of 1 µL onto the LC column. The overall run time was 7 min, including 3 min of post-run time. The column effluent was introduced to the mass spectrometer and monitored. The LC–MS/MS system composed of the Agilent 1290 infinity UHPLC, coupled with Agilent 6410 Triple quad LC/MS. The Agilent MassHunted Qualitative Analysis for QQQ B.04.00/Build 4.0.225.19 software package (Agilent, Santa Clara, CA, USA) was used for the acquisition and processing of data.

Tandem mass spectrometry (MS/MS) was performed on the positive ion electrospray ionization (ESI) mode. Collision-induced dissociation (CID) of the samples were carried out. The mass of the precursor ion was scanned in the 1st quadrupole (Q1), m/z selected and collision activated in Q2, and the products were analyzed in the third quadrupole (Q3). Mass transitions of m/z were optimized for TAMX and 4-OHT. The optimal MS settings were adjusted manually. The desolvation nitrogen gas was set at 11 L/min, while the ion spray voltage at 4 kV and its pressure was at 45 psi. The desolvation temperature was maintained at 350 °C. Acquisition was performed in the multiple reaction monitoring (MRM) mode.

### 2.9. Histopathology Studies

The organs (liver, lung, and brain) and tumor tissue harvested were fixed in 10% buffered formalin for 48 h, before processing for histopathological studies. The fixed tissues and organs were placed in cassettes and processed through various solvents (80% ethanol (2 h; 2 flasks), 95% ethanol (2 h; 2 flasks), 100% ethanol (3 h; 3 flasks), chloroform (3 h; 3 flasks), and paraffin wax (5 ½ h; 2 flasks)), using an automated tissue processor (Leica TP1020 Automatic Tissue Processor, Leica, Wetzlar, Germany). Then, the tissues were sectioned (5 µM) using the rotary microtome (Leica, Germany), and dried overnight in an oven set at 37 °C. The sections were stained with hematoxylin and eosin (H&E), according to the protocol by Harris and coded according to [32]. For each study group, sections from a minimum of three mice (*n* = 3) were examined for histopathological changes by a pathologist, who was blinded to the groups.

### 2.10. Statistical Analysis

All data were expressed as a mean ± standard deviation (*n* = 3). Significant difference in mean values were analyzed using one-way analysis of variance (ANOVA) with post-hoc Tukey, using GraphPad—a web-based program. A *p*-value of less than 0.05 (*p* < 0.05) was considered to be statistically significant.

## 3. Results and Discussion

### 3.1. Skin Irritancy Studies

It is essential to determine the skin irritancy potential of the formulations and its constituents prior to performing efficacy studies using an animal model. These studies would provide information, if the excipients (e.g., SPC, NaTC, the ratite oils, or the drug (TAMX or 4-OHT)) in the formulations would cause any irritancy on the skin of mice. In addition, it was also important to determine if shaving prior to application or skin abrasion, caused any irritations. For this purpose, the mice were subjected to skin irritancy studies. A defined area of the breast was treated daily with emu oil, unloaded transfersomes (SPC: EA, 85:15 (*w*/*v*)), or 4-OHT-loaded transfersomes (with emu oil) and the control (shaved mice), continuously for 10 days. The results from the skin irritancy experiment showed that none of the formulations induced any skin irritation over the 10 days of application (Figure 2A,B); with a normal Draize score of zero. Furthermore, histopathological analysis of skin biopsies from all four groups showed normal epidermis, with all layers of squamous epithelium intact, and there was no evidence of any discontinuity, ulceration, or blistering. The underlying dermis showed adequate vasculature and fibrous connective tissue stroma with scattered sebaceous glands and hair follicles. The dermo-epidermal junctions were normal with no evidence of inflammation or necrosis (Figure 2C,D). As such, the emu oil, transfersomal formulations, and its constituents were deemed safe and were used in subsequent experiments.

### 3.2. Treatment with Emu Oil and 4-OHT Transfersomal Formulations

Therapeutic effects of emu oil and transfersomal formulations (with/-out ratite oil) loaded with 4-OHT were investigated using three protocols, as described in the methods section. TAMX administered orally was used as the positive control. There was no difference (*p* > 0.05) in the mean tumor volume in animals treated with emu oil as compared to orally administered TAMX, up until day 9 (Figure 3). In addition, when tumor growth between animals treated with emu oil was compared against the untreated group, there was a marked difference (*p* < 0.05) in tumor size observed from day 7 onwards. This showed that emu oil possessed some potential anti-tumor activity, which might be comparable to TAMX, up until day 9. This anti-tumor effect exhibited by emu oil might be due to its high content of polyunsaturated fatty acids (PUFAs). Several studies found omega-3, 6 PUFAs; including eicopentaenoic acid (EPA) and docosahexaenoic acid (DHA) from emu oil to have anti-proliferation and cytotoxic effects on cultured cancer cells [35,36,37]. A substantial decrease (*p* < 0.05) in tumor volume was observed from day 14 onwards, in mice treated with both the 4OHT transfersome formulations, with and without emu oil (DLT–EO (85:15), and DLT, (85:15) as compared to the untreated mice (Figure 3). However, there were no significant difference (*p* > 0.05) in tumor volume in these mice, when compared to the oral TAMX group (*p* > 0.05). Interestingly, the formulations containing 4-OHT with and without emu oil showed no significant difference in tumor volume throughout the experimental period. A possible explanation could be that the presence of 4-OHT might mask the anti-tumor activity of emu oil, or DLT by itself served as an effective penetration enhancer. Furthermore, there might be antagonistic interactions between 4-OHT and emu oil. However, further studies need to be carried out to confirm this. These results provide evidence that transdermal application of the transferosome formulations could reduce tumor volume, which were comparable to the findings observed with orally administered TAMX, suggesting that these have the potential of being developed into effective topical anti-cancer agents.

Development of the method to quantify tamoxifen and its metabolite, 4-hydroxy tamoxifen in mice plasma.

Circulating concentrations of TAMX and its metabolites 4-OHT in blood plasma was quantified, after we developed and validated an LC–MS/MS method, as per the FDA guidelines. Method development began with optimization of the chromatographic conditions, including flow rate and injection volume. In order to optimize the ESI conditions for TAMX and 4-OHT, quadrupole full scans were carried out in positive ion detection mode. The mass spectra for TAMX and 4-OHT revealed peaks at *m*/*z* 372.5 and 388.2, respectively, as protonated molecular ions [M + H]^+^ (Appendix A). Following detailed optimization of the mass spectrometry conditions, *m*/*z* 372.2 → 70.1 and 388.2 → 72.0 at low resolution for TAMX and 4-OHT, respectively, was used for the quantitation purpose. The LC–MS/MS method was developed with a one-step sample processing and no derivatization allowed a simultaneous analysis of TAMX and 4-OHT. The chromatograms (Appendix A) compared three different concentrations (200, 20, and 2 ng/mL) of TAMX and 4-OHT spiked into the mice plasma. It was evident that the developed method had a high specificity and selectivity in the determination of TAMX and 4-OHT, without causing any interference in mice plasma. The LC–MS/MS method was validated for specificity, linearity, precision, and accuracy. The calibration curves were constructed using five different concentrations of TAMX and 4-OHT. Linear regression and ANOVA statistical data of TAMX and 4-OHT are presented in Table 1. The average correlation coefficient was found to be 0.994 and 0.996 for the TAMX and 4-OHT, respectively. Correlation coefficients (*r*) of above 0.99 indicated a good linearity with the drug [38]. The limit of detection (LOD) and quantification limit (LOQ) was determined using the standard deviation (S.D) of the response slope, as recommended by the ICH guidelines [39]. The distribution variable (*F*) for lack of fit was smaller than the tabulated *F* value for 95% confidence, hence, the LC–MS/MS showed no lack of fit.

Precision and accuracy assessed by analyzing the QC samples of low, medium, and high concentrations, deemed the method to be repeatable and accurate (Table 2). The low % relative standard deviation (RSD) values of repeatability for inter-day (0.5–2.35%) and intra-day (0.50–2.37%) variations of both compounds, established that the proposed method was precise (Table 2). These results, therefore, demonstrated the methods’ reproducibility, as the introduced variations in the test had no influence on the experimental results. Accuracy ranging from 93.9 to 108.0% (Table 2), was found to be within the limits of acceptance for bioanalytical methods, as per the FDA guidelines [34]. This method was, therefore, suitable for drug monitoring at low doses of TAMX and 4-OHT regimens, which require selective and sensitive techniques, particularly for biological samples.

#### Quantification of TAMX and 4-OHT in the BALB/c Mice Plasma

The established LC–MS/MS method was applied to determine the amount of TAMX and 4-OHT in the plasma obtained from mice that were fed daily with 0.6 mg/kg/day of TAMX (positive control) or subjected to topical application of transfersome formulation (1 mg/mL/day (100 µL of 4-OHT). Concentrations (mean ± S.D) of TAMX and 4-OHT in plasma obtained 18-days post treatment in samples taken 24 h post administration is shown in Table 3. Plasma concentration of 4-OHT in mice fed with TAMX was found to be 2-folds higher than TAMX. This could be due to the way the TAMX was metabolized, causing accumulation of 4-OHT or lack of its clearance. A previous study had reported that although clearance of TAMX from mice sera was faster, as compared to human sera; accumulation of TAMX and its metabolites in tissues were found to be significantly elevated, 24 h after daily doses of TAMX (40 mg/kg) for 5-days [40]. In addition, some studies found the 4-OHT/TAMX ratio in mice to be opposite to that observed in rats and human [41,42,43]. Topical application of the transferosome formulations produced a slight increase (0.03% and 0.01%) in accumulated 4-OHT, in the plasma, when administered via DLT 85:15 (0.03%) and DLT–EO, 85:15 (0.01%). The findings suggest that transfersome formulations without emu oil had a 3-fold increase in plasma 4-OHT; indicating that emu oil might inhibit uptake of 4-OHT into the plasma. These results support our earlier ex vivo permeability studies on porcine skin using Franz diffusion cell, where we reported reduced permeation of 4-OHT in formulations with emu oil [27]. However, despite the lower plasma levels of 4-OHT, the formulation with emu oil showed a similar reduction in tumor volume, as compared to the formulation without emu oil (Figure 2).

The daily dose of TAMX (0.12 mg) and 4-OHT (0.1 mg) administered were almost similar, but the percentage of 4-OHT accumulating in plasma in animal fed with TAMX was (0.53%) (*p* < 0.05), compared to animals that received transdermal applications (Table 3). In addition, topical administration of 4-OHT via the transfersomes (DLT and DLT–EO) showed marked reduction (*p* < 0.0001) of plasma 4-OHT, compared to the oral route (Table 3). Interestingly, the transfersome formulation with emu oil (DLT–EO) showed a significantly lower (*p* < 0.0001) systemic 4-OHT accumulation, compared to the formulation without emu oil. This could also be due to 4-OHT being trapped by the emu oil at the site of the tumor. Hence, preventing the absorption of 4-OHT from the tumor cells into systemic circulation. However, if this was to be true, the formulations with emu oil should have shown an enhanced antitumor effect, compared to the formulation without emu oil. However, as observed in the earlier studies (Figure 2), no significant difference in mean tumor volume between these formulations were observed. As mentioned previously, a possible reason could be due to the antagonistic interaction between emu oil and 4-OHT. Whilst topical formulations displayed lower amounts of 4-OHT in the plasma, their effects in reducing tumor size was found to be comparable to the orally administered TAMX (Figure 2). Similar findings were reported in previous studies comparing the efficacy of oral TAMX with 4-OHT hydro-alcoholic gel, applied topically in humans [44,45,46]. An extensive systematic review and meta-analysis of these studies reported that plasma levels of 4-OHT was a significantly lower in patients who were treated with topical 4-OHT hydro-alcoholic gel, compared to oral TAMX [16].

Circulation of TAMX or 4-OHT was reported to be detrimental to humans, as this could result in various side-effects like uterine [47] and endometrial cancers [48]. Hence, reduced levels of plasma 4-OHT could be taken as an indication that these transfersome formulations were potentially safer to use. Despite the cost of 4-OHT being higher than TAMX, this approach might be safer for long-term use. In addition, the amount of 4-OHT in topical formulations were rather small.

### 3.3. Histopathology Studies

There was no evidence of tumor metastasis in the lung or brain of animals from all five groups (Figure 4). In addition, microscopic observations of lungs showed normal alveoli with mild scattered non-specific inflammatory cells. We found extensive areas of metastatic deposits around the central vein and in between the hepatic parenchyma in liver sections from the control group (Figure 5A). In the liver sections of mice fed with TAMX, we observed many dilated central veins and fewer scattered micro metastases in the parenchyma, when compared to the sections from the control group, indicating a reduced metastasis (Figure 5B). Liver sections from animals treated with pure emu oil or the 4-OHT transfersome formulation, with or without emu oil (Figure 5C–E), showed fewer metastatic deposits that were scattered, compared to the sections from the oral TAMX and the control groups. Some studies reported that the in situ tumor that grow in mice inoculated with 4T1 cells normally metastasize to liver and lungs, as early as 8-days post inoculation [49,50]. Previous studies found lungs to be the first site where metastasis can be detected, followed by bone and liver [51]. Metastases refers to the development of secondary malignant growth at a distance site from the primary site of cancer, which happens when the tumor cells break away from the primary site and get into the circulatory or lymphatic systems, permitting these cells to migrate to distant sites. Micro-metastases and circulating cancer cells in blood, in bone marrow, and in the lymph nodes are almost constant in advanced cancerous disease [52] and in 95% of early stage I/II breast cancers [53,54].

Tumor section from the control group showed poorly circumscribed carcinoma with diffused sheets of malignant cells in the periphery and there were some diffused areas of necrosis in the center of the tumors (Figure 6A). In addition, the tumor tissue showed abundant vascularization, with some invasion in to the chest wall muscles. The tumor sections from mice fed with TAMX showed necrotic areas on the tumor, indicating dead tumor tissues (Figure 6B). Similar features, i.e., large necrotic areas, were also seen on sections from mice treated with pure emu oil (Figure 6C) or 4-OHT transfersomal formulations, with (Figure 6D) or without (Figure 6E) emu oil. The presence of large necrotic areas in the tumor sections from mice that were treated with topical applications containing 4-OHT groups (emu oil, DLT, and DLT–EO (85:15)) might indicate that many tumor cells died and there were lower number of cells that were able to metastasize. This could explain why there were fewer scattered metastases in the liver. This was in contrast to what was observed in tumor sections from the TAMX-fed animals, where the necrotic areas were smaller and a higher level of metastasis in the liver was observed. The results from tumor volume analysis did not reflect how oral TAMX and topical formulations of 4-OHT used in this study affected tumors. Tumor volume measurements on its own might not provide independent prognostic information, beyond the standard pathological parameters [55,56]. However, as in this study, tumor volume measurement, supported by histopathological studies, can give a more accurate description on the effectiveness of the treatment method.

## 4. Conclusions

Skin irritancy study showed that the pure emu oil and transfersome formulations, with or without the emu oil, did not cause skin irritancy in the animals studied. In terms of tumor size reduction, 4-OHT transfersome formulations with and without emu oil gave comparative results with oral TAMX. There were lower levels of 4-OHT in the plasma of animals subjected to topical treatments, which was an indication that delivering this drug via this route could avoid systemic circulation, and this in turn could reduce distribution of this drug to tissues that are susceptible to TAMX or 4-OHT induced toxicity. A higher degree of necrosis was observed in the tumor sections of animals that received topically administered 4-OHT, which further strengthens the finding that these formulations might be more effective than orally administered TAMX. Despite having reduced plasma 4-OHT concentrations 4-OHT formulations administered transdermally displayed treatment efficacy that were comparable to the oral TAMX. However, the quantification of 4-OHT and TAMX concentrations in tumor tissues would also provide more information in this regard. This is the first study that reports a lowered systemic circulation of 4-OHT, with the use of emu oil in a local transdermal formulation, for the treatment of breast cancer. Further studies are required to understand this role of emu oil for its future use as a local transdermal therapy. 

## Figures and Tables

**Figure 1 pharmaceutics-12-00807-f001:**
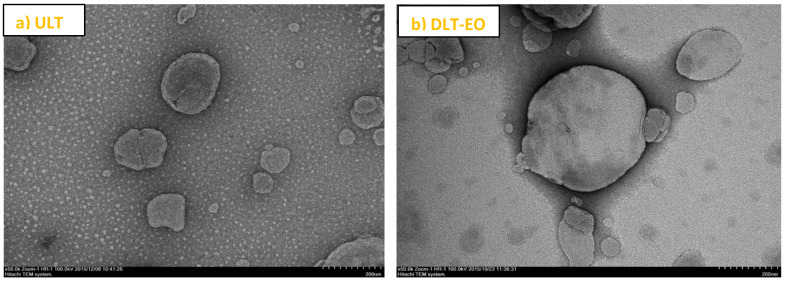
TEM images of transfersomes prepared in this study. The images approximately represented the morphology of spherical liposome vesicles. (**a**) ULT—unloaded transfersomes; and (**b**) DLT–EO—drug loaded transfersomes with emu oil at magnifications of ×50.0 k. (adapted from [27]).

**Figure 2 pharmaceutics-12-00807-f002:**
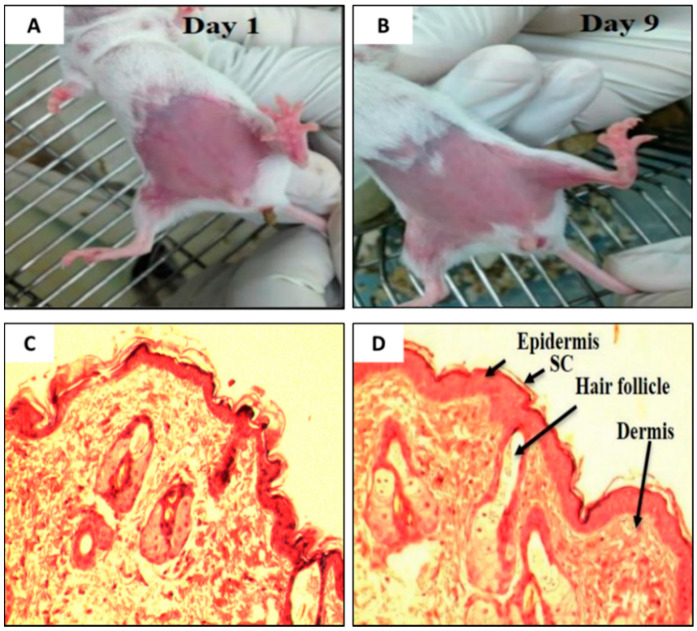
Visual inspection on the BALB/c mice on (**A**) Day 1 and (**B**) Day 9. (**C**) Photomicrograph of the dermo-epidermal junctions of mouse skin treated with (**C**) Day 1 (Control) and (**D**) Day 9 (Drug loaded transfersome with emu oil, DLT–EO), (H&E, 100×). (Note: As the results were the same across all groups of mice, the pictures and photomicrograph of only two group is shown above).

**Figure 3 pharmaceutics-12-00807-f003:**
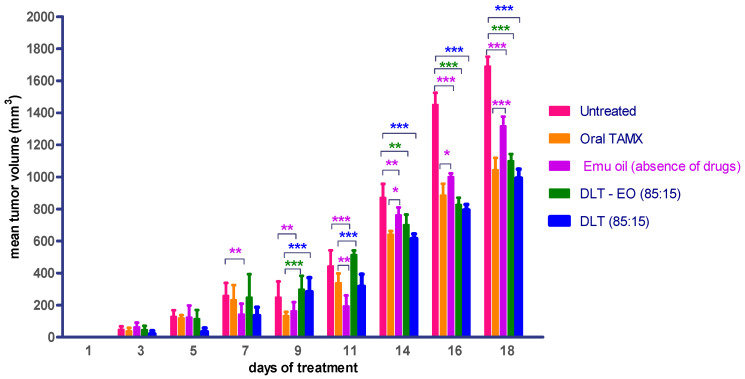
Mean tumor volume of BALB/c mice treated with oral TAMX, emu oil, and transfersomal formulations with/without emu oil. DLT—deformable liposomes, DLT–EO—deformable liposomes with emu oil. Each value is the mean ± SD (*n* = 3). (* *p* < 0.05; ** *p*< 0.01; *** *p* < 0.001).

**Figure 4 pharmaceutics-12-00807-f004:**
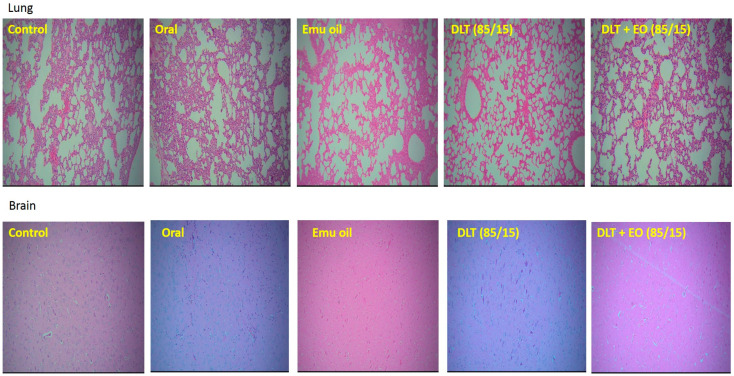
Photomicrographs from the lung and brain of the control (untreated), orally treated (TAMX), topically treated with emu oil (without the presence of drugs), formulations; DLT–EO (85:15) at day 18 post treatment. The lung showing normal alveoli with mild scattered non-specific inflammatory cells (H&E, 100×), in normal brain (H&E, 100×). No evidence of tumor metastasis in both the lung and brain of all groups (treated and untreated) were observed. Note: DLT—deformable liposomes, DLT–EO—deformable liposomes with emu oil.

**Figure 5 pharmaceutics-12-00807-f005:**
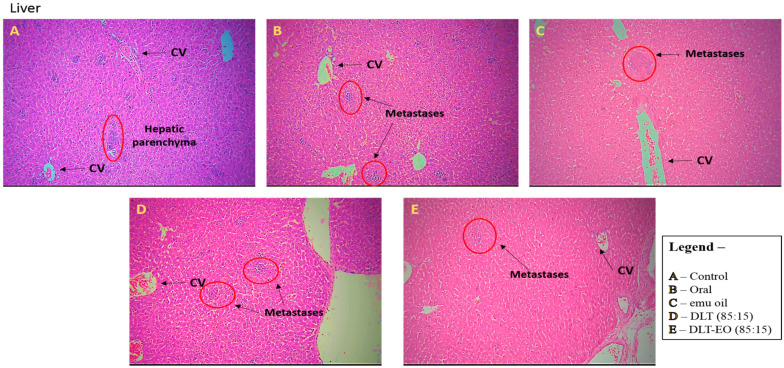
Photomicrographs from the liver of the (**A**): control (untreated), (**B**): orally treated (TAMX), (**C**): topically treated with emu oil (without the presence of drugs), formulations; (**D**): DLT (85:15) and (**E**): DLT–EO (85:15) at day 18 post treatment. H&E magnification 100× Note: DLT—deformable liposomes, DLT–EO—deformable liposomes with emu oil, CV—central vein.

**Figure 6 pharmaceutics-12-00807-f006:**
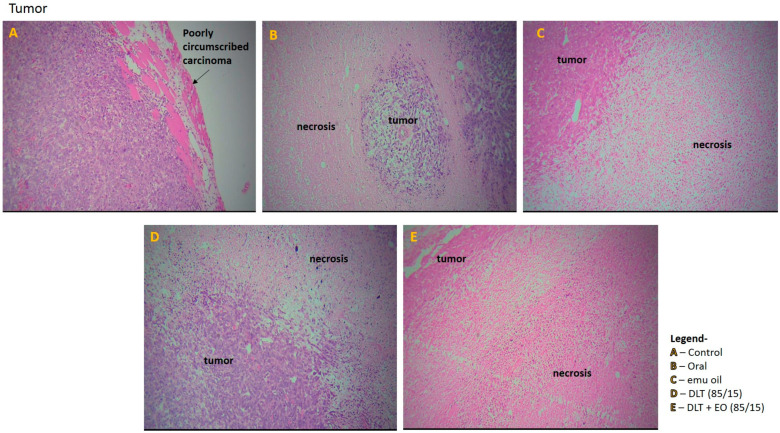
Photomicrographs from the tumor of the (**A**): control (untreated), (**B**): orally treated (TAMX), (**C**): topically treated with emu oil (without the presence of drugs), formulations; (**D**): DLT (85/15) and (**E**): DLT + EO (85/15), at day 18 post treatment. H&E magnification 100×. Note: DLT—deformable liposomes, DLT + EO—deformable liposomes with emu oil.

**Table 1 pharmaceutics-12-00807-t001:** Linear regression and ANOVA statistical data in the analysis of TAMX and 4-OHT.

Statistical Parameters	TAMX ^a^	4-OHT ^a^
Concentration range (ng mL^−1^)	50–0.5	50–0.5
Regression equation	*y* = 0.8144*x* + 7.116	*y* = 1.6851*x* + 1.705
Correlation equation (*r*)	0.9935	0.9956
Limit of detection (LOD) (ng mL^−1^)	0.1	0.1
Limit of quantification (LOQ) (ng mL^−1^)	0.3	0.3
Standard error	5.74	2.20
*F*	3974.99	1120.60
*SS* (residual)	263.66	24.11
*MS* (residual)	32.96	4.82
*SS* (regression)	5404.55	1.31 × 10^5^
*MS* (regression)	5404.55	1.31 × 10^5^
Lower 95%	0.98167	3.960
Upper 95%	4.3916	13.696

^a^ Each value is the mean ± SD (*n* = 3).

**Table 2 pharmaceutics-12-00807-t002:** Intra-day and inter-day precision and accuracy of TAMX and 4-OHT.

Compounds	Quality Control	Nominal Concentration (ng·mL^−1^)	Precision (% RSD) ^a^		Accuracy (%) ^a^
			Intra-Assay	Inter-Assay	
TAMX	QCL	1	2.03	2.01	96.7
	QCM	20	0.67	0.67	93.9
	QCH	200	2.37	2.35	107.1
4-OHT	QCL	1	1.71	1.69	95.5
	QCM	20	0.50	0.50	95.7
	QCH	200	1.48	1.47	108.0

^a^ Each value is the mean ± SD (*n* = 6). Abbreviations: QCL—quality control low; QCM—quality control medium; and QCH—quality control high.

**Table 3 pharmaceutics-12-00807-t003:** Plasma levels of TAMX and 4-OHT in BALB/c mice when comparing the two different routes of administration.

Types of Formulations	Dose Administered (µg)	Tamoxifen (ng/mL)	% of TAMX Compared to Dose Administered	4-OHT (ng/mL)	% of 4-OHT Compared to Dose Administered
Control	ND	-	-	-	-
Topical Pure EO	-	-	-	-	-
Oral TAMX	120	325.21 ± 3.78	0.27	634.42 ± 7.54	0.53
DLT (85:15) + 4-OHT Transfersome	100	BQL	-	32.45 ± 0.48 ***	0.03
DLT + EO (85:15) + 4-OHT Transfersome	100	BQL	-	10.24 ± 0.07 ***	0.01

*** *p* < 0.0001, significant value against oral TAMX and between DLT (85:15) + 4-OHT and DLT + EO (85:15) + 4-OHT. Values are the mean ± SD (*n* = 6). BQL—below the limit of quantification; EO—emu oil; DLT—deformable liposomes; and DLT + EO—deformable liposomes with emu oil.

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
