# Peer review of "Efficacy of Emu Oil Transfersomes for Local Transdermal Delivery of 4-OH Tamoxifen in the Treatment of Breast Cancer"

_pharmaceutics, 2020, doi:10.3390/pharmaceutics12090807_

Round 1

Reviewer 1 Report

Sundralingam et al. reported the biological evaluation of 4OHT-loaded transfersomes for breast cancer prevention. This topic is interesting and relevant, but several aspects of the manuscript require clarification prior to further consideration. Below, specific comments are summarized.

- Introduction: “ability to permeate through the pores on skin”. This is a very controversial topic; to support this statement, the authors must cite references that more clearly demonstrate that transfersomes permeate intact across cutaneous layers.

-  I suggest giving a brief background on emu oil and its biological effects related to cancer in the introduction, so that readers can appreciate its relevance. Additionally, it is necessary to provide its source and certification of its contents.

- In the irritation assay, the authors cited ostrich or emu oils. Were both used? Why?

- Transfersome formulation:  please describe the method of obtainment. Citing a submitted manuscript is not sufficient. How much drug was incorporated? What was the encapsulation efficiency? Was any “free drug” removed?

- Figure 1C: please include size bars. Even though the authors mentioned similarity, I suggest showing histological images of the skin of control animal (at least) and animals treated with emu oil to enable comparison by the readers.

-  Figure 2: groups treated with Emu oil and DLT are represented by colors that are too similar. I suggest altering one of the colors.

- In vivo efficacy: the authors stated that “This shows that emu oil possessed some potential anti-tumour activity, which might be comparable to TAMX”, however, they showed a significant difference after day 11; therefore, I suggest rephrasing this result. Overall, I believe this section deserves a richer discussion considering (i) the high cost of 4OHT, (ii) that emu oil reduced the tumor volume compared to untreated animals during the whole treatment, and (iii) that there was no potentiation of antitumour effects when emu oil and 4OHT were co-encapsulated. Moreover, to what reasons do the authors attribute the lack of potentiation when 4OHT and emu oil were co-encapsulated? Were there differences in the number of tumors or only in the dimensions?

- Figure 3: I found it very hard to compare the chromatograms on Figure 3 because the scale was not the same. I suggest using the same scale to compare the blank and various drug concentrations.

- Table 3 shows that 4OHT plasma levels when the animals were treated with DLT were smaller than with DLT+emu. Was this difference significant? In other words, does emu presence reduce cutaneous delivery of 4OHT? A smaller 4OHT penetration might help to explain why treatment with emu+4OHT did not result in a more pronounced antitumor effect compared to 4OHT alone.

Minor comments:

- please correct the redundancy in the following sentences: “Oral administration of TAMX in peanut oil (Group B) was administered using a plastic gavage tube” and “in the mean tumour volume of tumours from animals”

- 2.8. Centrifugation was performed at 13.2 rpm? What type of centrifuge was used? if possible, please report in g.

- Figure 5: please correct figure caption (it says: Figure Error! No text of specified style in document).

Reviewer 2 Report

This is  very good manuscript describing the new approach of targeting solid tumors, specifically, the breast cancer using local transdermal therapy.

The manuscript is rich in important information and the experiments are discussed in details.

I recommend publishing the paper after performing the following minor corrections:

The used approach of treating breast cancer using the local transdermal delivery has been previously reported in several studies that should be mentioned in the introduction such as:

Colloids Surf B Biointerfaces. 2018 Jul 1;167:63-72. 

Colloids Surf B Biointerfaces. 2017 Jul 1;155:512-521. 

Safwat S, Ishak RAH,Hathout RM, Mortada ND.

Line 81: Sodium Taurocholate. Please correct through the whole text.

Line 266: Please remove the letter “a”

Usually the LOQ is 3 times the LOD. Please comment on the obtained values.

Round 2

Reviewer 1 Report

Overall, the authors provided satisfactory replies to the majority my comments, but I must I insist in two of them:

- Figure 3 is not well prepared or presented. If the authors are unwilling to provide a better figure, I suggest including it as supplementary figure.

- Figure 1 needs an error bar.

Additionally, as to the source of the oil, my suggestion referred to adding few words in the methods to briefly outline the components/purity of the oil. No need to provide certificates.
